# Predicting Feature Imputability in the Absence of Ground Truth

**Niamh McCombe** [1]  **Xuemei Ding** [1]  **Girijesh Prasad** [1]  **David P. Finn** [2]  **Stephen Todd** [3]  **Paula L. McClean** [4]
**KongFatt Wong-Lin** [1]

## Abstract

Data imputation is the most popular method of dealing with missing values, but in most real life applications, large missing data can occur and it is difficult or impossible to evaluate whether data has been imputed accurately (lack of ground truth). This paper addresses these issues by proposing an effective and simple principal component based method for determining whether individual data features can be accurately imputed - feature imputability. In particular, we establish a strong linear relationship between principal component loadings and feature imputability, even in the presence of extreme missingness and lack of ground truth. This work will have important implications in practical data imputation strategies.

## 1. Introduction

Data imputation (replacing missing values with estimated values) is used for dealing with missing data. Appropriate data imputation approaches are necessary to ensure reliable and robust model classification performances (Friedjungová et al., 2019). There is a large literature evaluating the imputation and classification accuracy of various imputation methods. Almost all of this literature consists of simulation studies where synthetic missing data is introduced into originally complete data, and imputation accuracy over the whole dataset is evaluated against the known ground truth - for example (Zhang; Waljee et al.; Malarvizhi & Thanamani; Baneshi & Talei, 2012; Srivastava & Dolatabadi)

---
[1]Intelligent Systems Research Centre, Ulster University, Magee Campus, Derry∼Londonderry, Northern Ireland, UK [2]Pharmacology and Therapeutics, School of Medicine, National University of Ireland Galway, Galway, Ireland [3]Altnagelvin Area Hospital, Western Health and Social Care Trust [4]Northern Ireland Centre for Stratified Medicine, Biomedical Sciences Research Institute, Ulster University, Derry∼Londonderry, Northern Ireland, UK. Correspondence to: Niamh McCombe <mccombe-n@ulster.ac.uk>, KongFatt Wong-Lin < k.wong-lin@ulster.ac.uk>.

*Presented at the first Workshop on the Art of Learning with Missing Values (Artemiss) hosted by the $37^{th}$ International Conference on Machine Learning (ICML).* Copyright 2020 by the author(s).

However, with large and heterogeneous data, it is often not clear what data features (variables) should be considered for data imputation - the problem of feature imputability. (Saar-Tsechansky & Provost, 2007) introduces and explores the concept of feature imputability, which is the degree to which any feature can be imputed as a function of the other features in a dataset. However there is a surprising lack of literature exploring feature imputability in simulation studies. In particular, there is no work that actually allows efficient prediction regarding feature imputability when the ground truth is unknown.

This work addresses the above issue by proposing the use of a simple yet efficient algorithm, nonlinear iterative partial least squares (NIPALS) (Wold, 1975), to evaluate feature imputability even when the proportion of missing data is large. NIPALS was developed for conducting principal components analysis (PCA) in the presence of missing data. As a case study, we evaluate data imputation accuracy and feature imputability using an open dementia dataset in which a very large amount of missing data is synthetically introduced, mimicking the missingness pattern in clinical data. We found that there is a linear relationship between feature imputability and principal component loadings computed by NIPALS.

## 2. Materials and Methods

### 2.1. Data

The data for analysis was extracted from the ADNIMERGE table from the Alzheimers Disease Neuroimaging Initiative (ADNI)merge R package, which amalgamates several key tables from the ADNI open source dementia data (adni.loni.usc.edu). The ADNI open database included clinical and neuropsychological assessments with diagnosis labelled as healthy, mild cognitive impairment (MCI) and early Alzheimer's Disease (AD).

#### 2.1.1. FEATURE SELECTION TO REDUCE DATA SIZE

Feature selection was performed on the ADNIMERGE table using the information gain (IG) algorithm (Battiti, 1994). IG of a given feature is the reduction in disorder of the class variable, when the class variable is separated accord-

ing to that feature. We used the IG implementation in the FSelector R package (Romanski et al., 2018). Feature selection was used here to reduce the large size of the original dataset. Multivariate feature selection which optimises for orthogonality among selected features was not used as we wish to simulate real world clinical data which may have many highly correlated features. The 8 Cognitive and Functional Assessments (CFAs) which had the highest IG with respect to CDR-SB (clinical dementia rating - sum of boxes, an objective measurement of disease severity - see (Ding et al., 2018) were selected). Gender and Age were also included in this base dataset. The variables selected included subscales of the Everyday Cognition scale (Farias et al., 2008) (Ecog*), the Montreal Cognitive Assessment (Nasreddine) (MOCA), and Logical Memory - Delayed Recall from the Weschler Memory Scale (Weschler & Stone, 1997) (LDELTOTAL).

### 2.1.2. INTRODUCTION OF MISSING VALUES

We mimicked a missingness pattern observed in data from our local memory clinic. Missing values were synthetically introduced into the CFA variables in the base dataset. No missingness was introduced into Gender or Age. The missingness introduced was the MAR (missing at random) type, increasing with disease severity with a formula $P_{\texttt{miss}} = 0.48 \pm (0.06 * \texttt{MMSE})$, where $P_{\texttt{miss}}$ is the probability of any given value being missing, and MMSE was the normalised Mini Mental State Examination (Molloy & Standish, 1997) score in ADNIMERGE. MMSE was used in the formula due to its common use in both clinical and open datasets. The 0.48 was implemented to provide 48% missingness among the CFA variables. In total, 10 synthetic datasets with different random missing patterns were generated, to ensure robustness in the results.

### 2.2. Analysis

#### 2.2.1. PRINCIPAL COMPONENTS ANALYSIS

We performed PCA on the base dataset using the princomp command built in to the stats package in R (R Core Team, 2019). Correlation (not covariance) method was used. Number of principal components was not specified in advance.

#### 2.2.2. MISSING DATA IMPUTATION

We used various algorithms to impute the synthetic datasets.

- Mean imputation - imputation of column mean, a computationally simple baseline.
- Median imputation - imputation of column median, as above.
- Predictive mean matching (PMM) from the multivariate imputation via chained equations (MICE) package

in R (Buuren & Groothuis-Oorshoorn, 2010). PMM is the default method for MICE, the most commonly used multiple imputation package. It is a multiple imputation method and we used the mean of 15 PMM outputs to calculate imputation accuracy. PMM takes a random draw from the posterior predictive distribution of the coefficients of a regression of observed values for each variable x on the other variables, to produce a new set of coefficients. These are used to predict x for both missing and observed values. Multiple imputations for each missing x are taken from cases with observed x where predicted x is close in value.

- missForest imputation (Stekhoven & Buehlmann, 2012) from the missForest package (Stekhoven, 2013) which is a popular iterative imputation method using Random Forest (RF)(Breiman) models. missForest begins with mean imputation. An RF model using observed values in each column as the dependent variable and all the other columns in the dataset as independent variables is built to impute missing values for each column in turn, until convergence.
- Probabilistic principal component analysis (PPCA) (Tipping & Bishop, 1999) is a probabilistic extension of principal component analysis, which uses a maximum likelihood(ML) approach to estimating the parameters of the latent variable model underlying the data. The probabilistic component allows for estimation of missing values. PCA can be seen as a special case of PPCA where the covariance of the error terms in the PPCA model is zero. An Expectation Maximisation (EM) iterative algorithm is used for ML estimation. We used the implementation in the PCAmethods (Stacklies et al., 2007) package in R with 3 principal components specified as determined by the kEstimate function.
- Bayesian principal component analysis (BPCA) (Nounou et al., 2002) s a computationally complex approach using Bayesian methods for PPCA component estimation. 3 principal components were specified. The implementation in the PCAmethods (Stacklies et al., 2007) package was used.
- Nonlinear iterative partial least squares (NIPALS) (Wold, 1975). NIPALS uses an alternating least squares algorithm to iteratively compute the scores and loadings of the first principal component (PC1), then PC1 is subtracted from the dataset and scores and loadings for the second principal component (PC2) are calculated, etc. NIPALS deals with missing values by using weighted regressions with missing values weighted at null. The implementation in the nipals (Wright, 2020) package was used; the number of principal components was not specified in advance.

The $R^2$ of the linear regression of the imputed values on ground truth (complete data) was used as a measure of im-

*Table 1.* Variable loadings on the first 3 principal components, and missForest feature imputability $R^2$

| VARIABLE | PC1 | PC2 | PC3 | $R^2$ |
|---|---|---|---|---|
| CDR-SB | 0.322 | 0.012 | 0.304 | n/a |
| Gender | 0.0719 | -0.679 | 0.195 | n/a |
| Age | 0.079 | -0.693 | -0.303 | n/a |
| EcogSPTotal | 0.390 | 0.071 | -0.194 | 0.862 |
| EcogSPMem | 0.368 | 0.045 | -0.068 | 0.821 |
| LDELTOTAL | -0.316 | 0.017 | -0.296 | 0.775 |
| EcogSPLang | 0.352 | 0.0350 | -0.148 | 0.763 |
| MOCA | -0.297 | 0.144 | -0.177 | 0.682 |
| EcogSPPlan | 0.356 | 0.103 | -0.285 | 0.797 |
| EcogSPVisspat | 0.346 | 0.123 | -0.306 | 0.791 |
| EcogPtTotal | 0.1959 | 0.0590 | 0.648 | 0.443 |

putation accuracy, with values ranging from 0 to 1 (poorest to highest in accuracy, respectively). The mean, minimum and maximum $R^2$ measurements from each of the 10 synthetic datasets were obtained. The average imputation $R^2$ of each individual variable using the missForest and PMM15 algorithms was also calculated.

### 2.2.3. REGRESSION ANALYSIS

The missForest and PMM15 imputation $R^2$ values for each CFA feature were regressed on the PC1 loadings of the full dataset with no missing values (calculated using the correlation method.) Further linear regression analyses was performed for each of the 10 datasets with synthetic missing values. missForest $R^2$ values for each feature were regressed on the PC1 loadings as calculated by the NIPALS method.

### 2.3. Software and Hardware

The above analyses and algorithms were run within R Studio version 1.146 on a Windows machine with R version 3.5.2(R Core Team, 2019) installed.

## 3. Results

### 3.1. Feature Selection and PCA Results

The 8 CFA features selected by IG and their loadings on the the first three principal components (PC1-PC3) are shown in Table 1. We find that most of the CFA variables selected (EcogSPTotal, EcogSPMem, LDELTotal, EcogSPLang, MOCA, EcogSPPlan, EcogSPVisspat) are loaded on PC1. PC2 is dominated by Gender and Age. The variable EcogPtTotal is loaded most strongly on PC3. The remaining principal components could be considered noise and are not shown in Table 1. This latent variable structure makes some intuitive sense as EcogPtTotal is the only selected Ecog* assessment which is completed by the patient; the others refer to study partner assessments.

### 3.2. PCA based imputation methods outperformed by RF and PMM

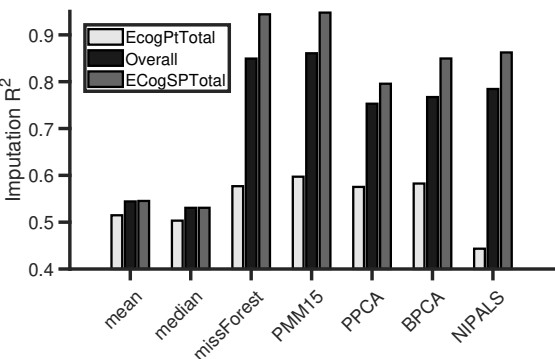

*Figure 1.* Imputation $R^2$ of imputation methods. Left-to-right groupings: mean, median, missForest, predictive mean matching average of 15 (PMM15), probabilistic principal component analysis (PPCA), Bayesian principal component analysis (BPCA), nonlinear iterative partial least squares (NIPALS). Least imputable feature imputation $R^2$, ECogPtTotal, light grey bars. Most imputable feature imputation $R^2$, ECogSPTotal, dark grey bars. Overall imputation $R^2$, black bars.

Using the synthetic missing datasets, we performed various imputation methods. We found that the Predictive Mean Matching (PMM) and Random Forest (RF) imputation methods provided the highest $R^2$ when tested against the complete dataset (ground truth) (Figure 1). We then investigate feature imputability by calculating the imputation $R^2$ of individual feature imputed values regressed against ground truth - feature imputability $R^2$ results for missForest are shown in Table 1, column 4 (Gender and Age, which are readily accessible in clinical data, and CD-RSB, the class variable, were not imputed). We find that missForest and PMM15 are the best performing imputation methods when measured against ground truth, outperforming all the PCA based methods. NIPALs is the best performing PCA based imputation method over the whole dataset. The feature imputability $R^2$ of the most and least imputable features (ECogSpTotal and EcogPtTotal) is also shown in Figure 1, in light grey and dark grey bars respectively.

### 3.3. Feature imputability highly correlated with principal component loadings

To explore the nature of feature imputability further, we hypothesise that feature imputability may be linked to the correlation of the features, as multivariate imputation algorithms predict each missing value as a function of the other features in the dataset (Stekhoven & Buehlmann, 2012; Buuren & Groothuis-Oudshoorn, 2010). Therefore we hypothesise that PCA methods may provide information about feature imputability. To explore this hypothesis we regress

feature imputability based on the best performing imputation methods (PMM15 and missForest) on the PC1 loadings (by correlation) of the complete dataset.

Figure 2 shows the relationship is almost exactly linear with $R^2 = 0.0.98 p = 1.5 \times 10^{-6}$ when imputation $R^2$ by variable of missForest(imp) is regressed on PC1 loadings(PC1). The resultant regression equation is $imp_x = 1.9 PC1_x + 0.19$ where x is a feature in the dataset. The regression using the imputation $R^2$ by feature of PMM15 imputation looks similar, with $R^2 = 0.987, p = 7.5 \times 10^{-7}$. The least imputable variable, ECogPtTotal, is furthest from the regression line, and more accurately imputed than the regression would predict.

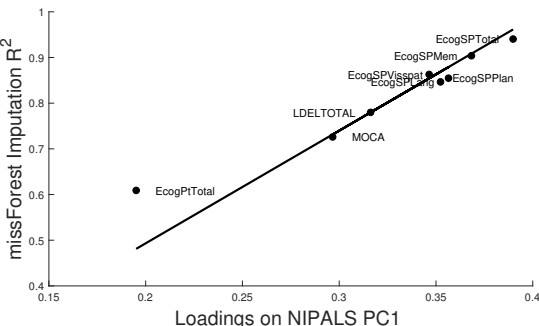

*Figure 3.* missForest $R^2$ by variable linearly regressed on NIPALS PC1 of a missing value dataset. $R^2 = 0.9116, p = 2.24 \times 10^{-4}$.

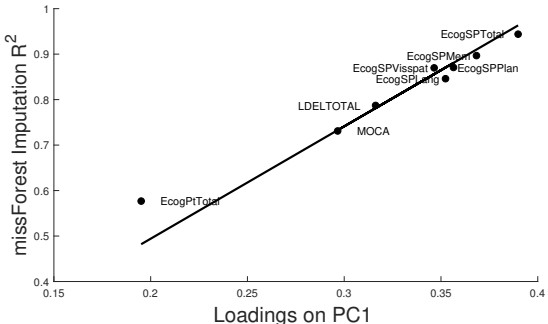

*Figure 2.* missForest imputation $R^2$ by variables linearly regressed on PC1 loadings of the complete dataset. $R^2 = 0.98, p = 1.5 \times 10^{-6}$

### 3.4. Strong relationship persists under extreme missingness and absence of ground truth

To determine whether it is possible to predict variable imputability even in the absence of ground truth and under conditions of large missing data, we use NIPALS, which performed the best out of the PCA based imputation methods, to calculate PC1 loadings for the missing synthetic datasets. For each dataset, we regress missForest imputation $R^2$ on PC1 loadings. The predictive power of PC1 loadings for variable imputability is still very strong, with $R^2$ for the synthetic datasets within range $0.8 - 0.95$. Once again the ECogPtTotal variable lies furthest from the regression line in all cases. An example regression, with $R^2 = 0.0.9116, p = 2.24 \times 10^{-4}$ is shown in Figure 3.

## 4. Discussion

We have found that feature imputability (Saar-Tsechansky & Provost, 2007) can be highly predicted from the principal component loadings on the dataset (Figure 2). As far as we know, this is the first time that such a relationship has been established. This strong relationship persists even when principal components are predicted from data with

extreme (48%) missingness (Figure 3). This means that even when the ground truth is not known, it is possible to predict with high accuracy which variables in this dataset can be accurately imputed. This simple yet accurate determinant of feature imputability could conveniently inform the decision on whether to impute a feature or omit it from a data model early in the pre-processing stage, and has the potential to inform further analysis of imputed datasets.

The logic underlying this strong relationship is that most of the correlations between variables in a dataset are captured by PC1, therefore loadings on PC1 explain how much a feature can be predicted from a basic linear model of other features in the dataset. Given that multivariate imputation methods impute each variable as a function of other variables in the dataset, PC1 may therefore provide an early glimpse of feature imputability. Where some features sit above the regression line, the imputation model for that feature has performed better than a simple linear combination of the other variables could achieve.

Future work will investigate different datasets, different types of missingness, and variations of current methods such as nonlinear PCA. We also plan to investigate strategies for handling less-imputable features, by designing data processing pipelines which specifically account for feature imputability. Overall, our work may potentially have important implications in practical data imputation strategies.

## Acknowledgements

This work was supported by the European Unions INTERREG VA Programme, managed by the Special EU Programmes Body (SEUPB) (Centre for Personalised Mediciane, IVA 5036)), and additional support by Alzheimers Research UK (XD, ST, PLM., KW-L), and Ulster University Research Challenge Fund (XD, ST, PLM, KW-L). The views and opinions expressed in this paper do not necessarily reflect those of the European Commission or the Special EU Programmes Body (SEUPB).

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
