# OpenReview forum: "Predicting Feature Imputability in the Absence of Ground Truth"
_ICML.cc/2020/Workshop/Artemiss — ICML Artemiss 2020_

### Official Review · AnonReviewer2 · 2020-06-18
**Predicting Feature Imputability in the Absence of Ground Truth**

**Rating:** 6
**Confidence:** 4

**Review:**

- There is better principal component methods than the NIPALS algorithm to impute with PCA (see the missMDA package).
- There is probably a bias in your methodology because your criterion is based on PC methods and you find that the best imputation methods are based on PC methods. Maybe if you consider a criterion based on regression, the imputation methods based on regression would be better. You should at least discuss this point.

---

### Official Review · AnonReviewer1 · 2020-06-23
**Nice idea; Methods section could be clearer; What actions can be taken with the output?**

**Confidence:** 3
**Rating:** 6

**Review:**

Summary: The authors propose a method to tackle the problem of feature imputability where the goal is to understand which features are most accurately imputable based on all other features in the data. This problem is interesting and aligns with the goals of this workshop. They show that using a PCA based imputation strategy, they can regress per-variable imputation performance on the first principal component loadings and this shows a linear relationship.

Strengths:
- The idea is interesting and deserves more exploration
- A number of commonly used imputation strategies are used

Weaknesses:
- The methods of the paper need to be made clearer (for example, there is no table with a ranking of which features are most or least imputable in the dataset, why was the proposed missingness simulation approach used, how is imputation performance measured without a gold standard etc.)
- It is not immediately clear to me how one would decide what to do after obtaining such a ranking of feature imputability, what constitutes a “low enough” score to justify removing the variable? Perhaps it is worth it to think about a hypothesis test for this?
- Only one dataset is used where PC1 contains a majority of the variance, this is not always the case for many data sources
- What happened to age and gender in the regression?

Further Questions:
- If a variable has low feature imputability, does that necessarily mean it should be removed? In the case that the missingness pattern is MCAR and the feature is valuable for predicting the outcome, removing it could degrade performance. Even if missingness is MNAR, it could still be worth preserving in a model to trade of predictive performance for poor inference.
- It may be worth thinking about nonlinear PCA and also test on datasets where many PCs are required to recover a good low rank approximation.


Overall, the authors present an interesting direction. The methodology of the paper could be made much clearer esp questions outlined above. More thought has to be put into actionable insights derived from the output of such a ranking approach.

---

### Decision · Program_Chairs · 2020-07-02

**Decision:**

Accept

**Comment:**

We are very happy to inform you that your paper has been accepted for the Artemiss workshop. We will contact you soon to inform you about the details concerning the format of your presentation at the workshop, and the camera-ready version deadline. Please take into account the referee's comments to write the camera-ready version.